# Recent Advances in Metal-Based Magnetic Composites as High-Efficiency Candidates for Ultrasound-Assisted Effects in Cancer Therapy

**DOI:** 10.3390/ijms221910461

**Published:** 2021-09-28

**Authors:** Zhenyu Wang, Xiaoxiao He, Shiyue Chen, Chengdian He, Teng Wang, Xiang Mao

**Affiliations:** 1State Key Laboratory of Ultrasound in Medicine and Engineering, College of Biomedical Engineering, Chongqing Medical University, Chongqing 400016, China; wangzhenyu@cqmu.edu.cn (Z.W.); hxx2240093537@163.com (X.H.); chen1321890343@163.com (S.C.); 15572071086@163.com (C.H.); wt15775806773@163.com (T.W.); 2Chongqing Key Laboratory of Biomedical Engineering, College of Biomedical Engineering, Chongqing Medical University, Chongqing 400016, China

**Keywords:** metal-based, magnetic materials, magnetic resonance imaging, ultrasound, cancer therapy

## Abstract

Metal-based magnetic materials have been used in different fields due to their particular physical or chemical properties. The original magnetic properties can be influenced by the composition of constituent metals. As utilized in different application fields, such as imaging monitoring, thermal treatment, and combined integration in cancer therapies, fabricated metal-based magnetic materials can be doped with target metal elements in research. Furthermore, there is one possible new trend in human activities and basic cancer treatment. As has appeared in characterizations such as magnetic resonance, catalytic performance, thermal efficiency, etc., structural information about the real morphology, size distribution, and composition play important roles in its further applications. In cancer studies, metal-based magnetic materials are considered one appropriate material because of their ability to penetrate biological tissues, interact with cellular components, and induce noxious effects. The disruptions of cytoskeletons, membranes, and the generation of reactive oxygen species (ROS) further influence the efficiency of metal-based magnetic materials in related applications. While combining with cancer cells, these magnetic materials are not only applied in imaging monitoring focus areas but also could give the exact area information in the cure process while integrating ultrasound treatment. Here, we provide an overview of metal-based magnetic materials of various types and then their real applications in the magnetic resonance imaging (MRI) field and cancer cell treatments. We will demonstrate advancements in using ultrasound fields co-worked with MRI or ROS approaches. Besides iron oxides, there is a super-family of heterogeneous magnetic materials used as magnetic agents, imaging materials, catalytic candidates in cell signaling and tissue imaging, and the expression of cancer cells and their high sensitivity to chemical, thermal, and mechanical stimuli. On the other hand, the interactions between magnetic candidates and cancer tissues may be used in drug delivery systems. The materials’ surface structure characteristics are introduced as drug loading substrates as much as possible. We emphasize that further research is required to fully characterize the mechanisms of underlying ultrasounds induced together, and their appropriate relevance for materials toxicology and biomedical applications.

## 1. Introduction

Magnetic materials have received tremendous attention in diverse fields because of their unique physicochemical properties and their potential utilization has widely expanded. In the past decade, a significant number of studies have focused on magnetic materials and their synthetic approaches. As appeared in previous works, these magnetic composites include: (1) Iron-based functional materials such as iron-based oxides and chalcogenides materials (Fe_3_O_4_, Fe_2_O_3_, FeO, FeS, FeS_2_, FeSe, FeSe, etc.), for which related magnetic or semiconductor properties among different applications have already been given [1,2]. According to the original synthetic approaches, iron oxides showed relative magnetic property can be adjusted by controlling the synthesis approaches during the crystalline growth processes. Also, the characteristics of iron oxides (size, structure, crystalline modification) were introduced to influenced magnetic characterization [3,4]. (2) As synthesized iron-based bimetallic nanocrystals (NCs) such as iron-based noble metal phased NCs (FeX, X = Au, Pt, Pd, Cu) and iron based non-noble metal NCs (FeX, X = Co, Ni, Mn) [5,6], they demonstrated clearly catalytic properties or biocompatibility in energy or biological applications. In this procedure, the magnetic property was mainly determined by the crystalline degree in the solvethermal method. (3) Heterogeneous magnetic composites were constructed through the use of magnetic particles as hard templates. The resultant construction was considered a two-isolated-parts integration and the function was merged as much as a multidisciplinary exhibition. According to material synthesis procedures, the metal-doped iron oxides (XFe_2_O_4_, X = Co, W, Ni) not only exhibit magnetic resonance in an applied field but also convey signal metal functional characteristics completely [7,8]. Attributed to magnetic resonance and metal physical properties, the attention on magnetic materials mediated the cross-pronouncing that occurred between magnetic agents and their biological applications. Especially in novel fields, it promoted the activation of signaling approaches and guided new types in cure applications [9,10,11]. Here, there were different choices. While the physical properties of magnetic agents are determined by their inorganic core, their surface properties also play an important role, especially in effectively interfacing (e.g., ensuring biocompatibility and specific localization) with biological systems, such as proteins, cells, and tissues [12,13]. It faced the NCs’ utilization for the biological application with magnetic and colloidal stability.

As in the utilization of dextran and polyethylene glycol (PEG), the merged versatile surface functionalization with antibodies, aptamer, peptides, and small molecules directed the magnetic agents to provide additional functions for multimodal imaging and therapies with target specificity [14]. Surely, the magnetic function was particularly being investigated for its biomedical applications as in magnetic resonance imaging (MRI) [15,16]. Eventually, the biorthogonal chemistry of magnetic agents broadened their applications to therapeutic areas [17]. It implied these magnetic agents can be rapidly taken up by the reticuloendothelial system (RES) and successfully used to detect tumor lesions in the liver. During whole constructions synthesis, the resultant magnetic agents can be treated as the hard templates in fabricating photosensitizer and sonosensitizer structures. Additionally, it illustrated the multifunctional characteristics in full utilization. By contrast, the smaller ones (≤50 nm) could escape phagocytosis to some extent with a prolonged circulation time [18]. It conveyed the size, this kind of importance gave the different exhibition in the biological applications, it was only one side in biological systems. In reference to magnetic materials, some of the modified inorganic shells or parts can expand the range of resultant magnetic nanocomposites due to each composition [19]. Combining with the ultrasound field showed the phenomenon expressed not only in MRI monitoring, but also conveyed in physical treatment in cancer therapies, without any contact functions [20,21]. Obviously, these therapeutic applications have shown new possibilities for the treatment of cancer. Due to the particularity of their structural construction, magnetic compositions exhibit hyperthermia in therapeutic cures, which can efficiently generate heat under remotely applied alternating magnetic fields in order to induce cancer cell death noninvasively—applied only toward the targeted area, with minimal damage to normal tissues. It is noted that the hyperthermia application of magnetic materials is not within the scope of this review, thus these were disregarded, as were modified forms of purified magnetic constructions. Magnetic composites can also be used for magnetic force-guided drug delivery with increased efficacy under different physical condition [22,23]. Here, we compiled research data from numerous publications, working on different aspects related to magnetic materials and applied fields as shown in Figure 1. As in the first part of this review, Fe_3_O_4_ as an effective agent in magnetic resonance imaging (MRI) utilization was discussed. Depending on the modification types and initial element precursors, these kinds of magnetic agents were tailored to turn the property of the final products into synthesis routes. Next, the occurrence of magnetic agents referred to the Fe_3_O_4_ based heterogeneous structure, with different structures, such as core-shell, core-satellites, and yolk-shell structures. This was treated as one effective supporter while ultrasounds were integrated in existing cancer therapies. Also, the synthesis of metal-doped X_Z_Fe_X_O_y_ materials was discussed while integrating ultrasound resonance into cancer therapies. It is noted that the reactions in iron elements linked with organic frameworks for forming magnetic metal–organic frameworks (MMOF) revealed a high possibility for further constructions of multifunctional metal-based magnetic composites. This could introduce the opportunity for loading drug or target molecules in MMOF, to be used as magnetic agents and drug carriers toward organ tissues. Additionally, these new types of magnetic composites can derive constructions as drug delivery vehicles in ultrasound fields for the treatment of cancer. The common approaches of synthesizing metal-based magnetic composites were outlined, including conventional chemical approaches, the physical path under extreme conditions, and the metal diffusion phenomenon through the etching method. The highlighted synthesis mechanisms were involved in the formation of metal deposition and alloyed diffusion. This review article intends to provide an authoritative reference for the numerous researchers working on various aspects of metal-based magnetic composites as a high-efficiency candidate for ultrasound-assisted effects in cancer therapies.

## 2. Superparamagnetic Iron Oxides Nanoparticles (SPIOs) as an Effective Agent in MRI Utilization

MRI was utilized based on very strong magnetic fields and electromagnetic alternating fields in the radio frequency with which certain atomic nuclei (usually hydrogen nuclei/protons) in the body can be resonantly excited; the receiver circuit in the electrical signals then collected it as data. The device was not harmful like X-rays or other machines where ionizing radiation is used. An essential basis for the image contrasts in MRI are the different release times of different tissue types. Also, the different contents of hydrogen atoms contribute to different tissues. Superparamagnetic iron oxides nanoparticles (SPIOs) can exhibit sensitive magnetic properties while under a magnetic field as much as possible. In order to determine the resonance while combined with different tissues’ engineering, the MRI effect showed distinguished signal contrast. Even current magnetic materials offered sufficient choices for full utilization in MRI applications, but the challenge was how to expanded and innovate original magnetic materials, how to improve the intensity and sensitivity in imaging processes, and how to achieve a lower dose and higher reflection. Among the imaging techniques have been improved, the conventional approaches in practical clinical applications such as the radiation exposure of computer tomography (CT) and position emission tomography (PET), as well as fluorescent agent imaging are well-utilized [24]. Due to high concerns about the aforementioned contents, magnetic resonance imaging (MRI) was expected to play an important role in clinical diagnosis. In addition, MRI contrast agents were the essential candidate for the main application area and gave the basic requirements in fabricating proper magnetic candidates. Compared with current magnetic materials, SPIOs were used as an imaging modality in the clinic, which had several advantages, such as good hydrophilicity, biocompatibility, anatomic detail, enhanced soft tissue contrast, high spatial resolution, etc. [25]. During the synthesis procedures, the surface modification and template-mediated heterogeneous structures as MRI candidates can also be demonstrated [26]. In order to obtain a magnetic resonance signal, the target tissues should be placed in strong magnetic fields, similar to nuclear magnetic resonance (NMR), in which the hydrogen nuclear spins are oriented in the same direction (low/higher-energy state). Here, the nucleus absorbs the energy and the number of spins with a higher-energy state increases while resonant radio frequency is irradiated.

According to previous studies [27,28], since the essential radio-frequency was deposited in real patients, there was unregulated absorption of the radio-frequency, which likely lead to increased energy release and temperature growth. These studies conveyed that iron oxides can be alternative T_1_ contrast agents due to their higher biocompatibility. Thus, conventional MRI was used with key parameters 1.5 and 3T MRI systems. Based on inherent tissue characteristics, T_1_ and T_2_ implied important exhibitions when SPIOs were utilized in MRI applications, as shown in Figure 2. Meanwhile, Fe_3_O_4_ can be degraded in the body and subsequently incorporated into iron pools or used in metabolic processes [29]. Fe_3_O_4_ could combine with particular metal ions for developing excellent compatibility in MRI. As is well known, T_1_ contrast is caused by the interaction between protons and nearby paramagnetic ions [30,31]. It has always used gadolinium ions combined with organic molecules in T_1_ exhibitions. Even this had several unpaired electrons, which gave the strongest contrast agent effect, but came with the toxicity of free ions, so metal–organism reagents were used conventionally [32]. So, it should be an appropriate choice to choose Fe_3_O_4_ in full, using T_1_/T_2_ contrast agents, due to its physical parameters such as relaxivity, magnetic signal intensity, and transverse relaxation [33,34]. Fe_3_O_4_ must have influences on MRI capacities during strong magnetic fields. For further investigation, the characteristics of the magnetic material itself should be controlled during synthesis or modification, due to its size, shape, and composition playing related roles in MRI applications. Superparamagnetic particles were used as T_1_/T_2_ contrast in real MRI applications. This can accelerate fast transverse (T_1_/T_2_) relaxation of hydrogen protons darkening the labeled region, which can act to increase the relaxation rate of hydrogen protons in their surrounding tissue water molecules, significantly enhancing the contrast of the targeted target region against the tissue background in order to improve imaging sensitivity. The size and shape should correspond to its magnetic property, and the influence factors not only convey excellent magnetic property, but also illustrate the penetrability in biocompatibility in real application. According to theoretical issues regarding the effect of particle size on relaxivity, three different regimes were given, along with size increasing, called the motional average regime (MAR), static dephasing regime (SDR), and echo limiting regime (ELR) [35,36]. As in the same argument, the magnetic property of Fe_3_O_4_ was affected by the distribution of iron ions in octahedral and tetrahedral sites of the spinel structure. The magnetic spins of the ions in each site were ferromagnetically coupled to each other and antiferromagnetically coupled with other sites. Therefore, the numbers of Fe^3+^ ions were the same in two different sites, and the magnetic spins cancel each other. Therefore, the magnetic spins of only Fe^2+^ ions in one site contributed to the net magnetic moment as much as possible.

The ordinary approach was illustrated in fabricating high-quality magnetic particles and carrying out surface modification in MRI applications. Among the synthesis steps, the particle was synthesized in the organic phase, and achieved high-uniformity size distribution. It did disperse in water mediums after the surface modifications [35]. There was one implied constant rule about the particle growth process as a crystalline- mediated growth process. The size, shape, and morphology of Fe_3_O_4_ were influenced while using organic molecules as the soft template, or the polarity of solvent was adjusted during heating processes [37]. Additionally, each of these characteristics could be adjusted for controlling the magnetic property, which directly reflected the positive relationship with increasing size, higher crystalline, and high stereospecific blockade. This involved physical resonance in MRI applications and might correspond well with the magnetic capacity of Fe_3_O_4_. Here, Fe_3_O_4_ could be fabricated in water mediums at normal temperature, but also could be fabricated in an organic solvent, and the resultant magnetic particles having a hydrophobic or hydrophilic surface, respectively. In the bio-transfer application, a scientific approach was presented, describing the fabrication of a nanoprobe that can act as a cardiac precursor label to segregate cells from cardiac/non-cardiac origins and be traced by MRI. During this, signal regulatory protein alpha (SIRPA) and kinase domain receptor (KDR) recognizing antibodies form a layer on the SPION-PEG complex and bind to a protein expressed on the surface of cardiac muscle cells. Physical attributes such as size, distribution, labeling efficiency, echocardiogram (ECG) changes, and bio-distribution by MRI were researched completely [38]. Along with the increasing attention to target tissue imaging and therapies, MRI technology can not only provide test-site anatomical information like computed tomography (CT) but has also provided us with tissues’ physiological and biochemical information. It is more sensitive than CT to distinguishing between normal or abnormal tissues, providing us with the correct organ function and physiological situation. According to the magnetic characteristics, the factors were mainly followed by size distribution, because of the increased crystalline-enhanced magnetization. It showed the lesion site clearly through the image scopes. Particularly in terms of lesion organ shape in functions before obvious changes, it should be of great help in the early detection of tumors and tumor identification. Surely, the development of the integration of diagnosis and treatment in target tissues requires new strategies towards cancer therapies. Therefore, novel approaches to applying MRI materials and ultrasounds in combined imaging characterization and treatment (physical and chemical methods) have become a most urgent demand in real applications.

## 3. Magnetic Particles Based Heterogeneous Structure as One Effective Supporter toward Ultrasound (US) Integrated Cancer Therapies

Therapeutic US was a physical method for the delivery of non-ionizing radiation in the form of mechanical sound waves into tissues to produce heat-releasing performance. The therapeutic effect of US depended on the dose (W. cm^2^ time) and dosage (frequency of application, series). It was usually exerted at two fixed frequencies of 1.0 MHz and 3.0 MHz and the most generally used deep-heating modality was able to attain depths of 5 cm and more below the surface of the body. In the illustration of US, similar to short-wave diathermy, it can be exerted in pulsed or continuous waves to apply therapeutic thermal and non-thermal efficacies. Therefore, the parameters of US technique depend on the desired effect and the density and location of the tissue under treatment. Similarly, using the cavitation effect of US in liquid mediums, the combination of magnetic materials and microbubbles to form composite materials, can not only realize the effect localization of MRI but can also use the cavitation effect to achieve the directional release of the drug delivery effect. This can combine the advantages of both and achieve the treatment of target cancer cells. During dual-model type application, combining US effect and MRI technique became an urgent requirement in mixed diagnosis or base treatment in different investigations. US treatment in imaging was based on the difference in US passage rates through tissue. As probes for US and MRI, magnetic agents should play two different roles while being utilized in tissue engineering investigations. They had totally different mechanisms in US procedures, leading to physical relaxation of the tissues surrounding the target tissues in the human body [39,40]. Compared with other imaging modalities, a single US imaging possessed many types of advantageous characteristics such as cost effectiveness, high safety, and portability [41], rendering it applicable to real-time imaging during operation as well as presurgical planning. However, US imaging was mainly used to determine superficial and echogenic structures, owing to its limitations in resolution and penetration. The sensitivity and resolution of clinical US imaging can be significantly improved with the utilization of contrast agents [42]. The particular characterization in the US process was the cavitation, which stabilized gas microbubbles as US contrast agents. These microbubbles typically consist of a lipid or albumin shell filled with an inert gas, which were used for blood pool imaging, because their relatively large size prevents them from leaking into the extravascular space [43]. In addition, microbubbles can be developed as molecular imaging probes via functionalization with targeting molecules such as antibodies and peptides [44]. Based on contrast agents, including liquid−liquid emulsions, gas−liquid emulsions, and solid NP, they have been reported to contribute to contrast enhancement in US imaging [45,46]. However, the multifunctional modal US and MRI agents were prepared via the incorporation of magnetic composites within the US contrast agent as much as possible [47]. For instance, some of the NP embedded bubbles were prepared through a one-pot synthetic strategy in emulsion polymerization of poly(butyl cyanoacrylate) along with oil-in-water encapsulation [48]. In these cases, the hybrid imaging agents showed strong contrast in US and MRI. The MRI effect can be enhanced after US-induced bubble destruction, demonstrating a triggerable MRI capability. In addition to magnetic particles themself, some of the chemotherapeutic drugs can also be efficiently incorporated into bubbles. This allows for multifunctional modal US–MR imaging and US-triggered drug delivery for primary tumor metastasis [49].

The additional real utilization, photoacoustic tomography (PAT), can create multiscale multi-contrast images of living biological structures ranging from organelles to organs. This emerging technology overcome the high degree of scattering of optical photons in biological tissues by making use of the photoacoustic effect. Light absorption by molecules creates a thermally induced pressure jump that launches ultrasonic waves, which are received by acoustic detectors to form images. Along with the development in fabricating hybrid magnetic materials as imaging agents with US and MRI response, it can be used in photoacoustic tomography (PAT), and is being developed by combining US and optical imaging [50,51]. The combination of two imaging methods (US and MRI) could overcome the limitations of optical imaging and ultrasound imaging. Since utilized organic molecules such as hemoglobin and melanin were used as intrinsic contrasts, PAT has been used to visualize various biological functions and tissue structures, including blood vessels, brain hemodynamics, tumor hypoxia, and the wound-healing process [52,53]. Furthermore, the contrast agents used can image deep tissues or transparent biological targets. These kinds of exogenous contrast agents for PAT include organic dyes, noble metal particles, carbon materials, and fluorescent organisms. Here, the main target of focus is how to merge MRI and PAT announcements together. Some of the multifunctional modal-probes for MRI and PAT were developed by combining magnetic and NIR-absorbing material [54]. A simple example such as Au@Fe_3_O_4_ core-shell structure was developed by reducing gold ions on the surface of Fe_3_O_4_ adsorbed with poly-L-histidine [55]. As shown in Figure 3, it was used as modal contrast in MRI and PAT eventually. This has illustrated one constant rule about NIP-absorbance materials being used in incorporation, not only referring to carbon materials, but also to metal-based magnetic composites. Some of the analyzed results for the merged properties fit of MRI, PAT, and US treatments have conveyed precisely this.

In addition, metal-based magnetic composites or heterogeneous structures have done well in exhibitions in the form of US treatments for cancer, as showed in Figure 4. Currently, the combination of magnetic materials with other functional materials has been well-investigated and used in cancer diagnosis and related treatments [56]. Especially in terms of a magnetic template as one coherent part, the characteristics of magnetic materials played important roles in making an integrated structure. For instance, nickel, iron, and cobalt elements could be used in the synthesis of dual-metal based magnetic composites and functional magnetic materials. In recent studies, a nickel ferrite/carbon nanocomposite (NiFe_2_O_4_/C) for combined multiple diagnostic and therapeutic functions, i.e., MRI, magnetic drug delivery, hyperthermia, and sonodynamic therapy (SDT) was synthesized [57]. Under ultrasound treatment (1.0 W. cm^2^), NiFe_2_O_4_/C could produce a large amount of ROS due to the catalytic metal composition (Ni, Fe). In addition, the excellent biocompatibility further expanded the application field of FeFe_2_O_4_, CoFe_2_O_4_, and NiFe_2_O_4_ in ultra-sensitive molecular imaging [58]. The biocompatibility of magnetic particle probes affected the detection sensitivity of biomolecules present in cancer cells. It was first examined whether these magnetic particles had any deleterious biological properties, and they was found to be biologically nontoxic in cytotoxicity studies on two different cell lines (HeLa and HepG2). Similarly, the ultra-smaller MnWO_X_-PEG can serve as a novel sonosensitizer for multi-modal imaging-guided SDT applications [59]. Due to its higher hydrophilicity in water mediums, its surface functionality should be well-understood in water-soluble or buffer solution dispersion. Amphiphilic polymers were used to synthesize MnWO_X_ and improve their water suspension stability in further measurements. The unique oxygen defect structure of MnWO_X_-PEG provided electron capture sites that can prevent the recombination of e^−^ and h^+^, resulting in a high ROS generation efficiency [60,61]. Moreover, MnWO_X_-PEG prevented glutathione (GSH) from clearing ROS, and further enhanced the effect of SDT treatment. The presence of Mn and W in this platform provided two diagnostic methods (magnetic resonance and computed tomography) that can be used to track the accumulation of materials in tumor applications [59,62]. Among the distinctive types of metal oxide NMs, manganese-based oxides, including manganese dioxide (MnO_2_), have been synthesized through appropriate approaches [63] and are of considerable importance in several technologies.

Manganese-based oxides have medical applications such as energy storage, imaging, theranostics, hyperthermia, drug delivery, biosensing, ultrasound-promoted tumor chemotherapy, and SDT. This kind of manganese-based structure has a tunable morphology and structure, inherent magnetic resonance, and a biodegradable character, with excellent biosecurity. It has the ability to degrade hydrogen peroxide in the cancer micro-environment to relieve hypoxia. During the full utilization processes, the oxygen generation as well as the produced ability of GSH depletion could enhance hemodynamic, PDT, SDT, and starvation therapies, and might provide favorable application in T_1_-MR imaging [64] and SPION, or oxidation-formed maghemite (γ-Fe_2_O_3_), with diameters ranging between 1 nm and 100 nm. The consisted iron oxide cores could be targeted to a specific location via externally applied magnets. A magnet was placed externally over the targeted area, producing a strong magnetic field that attracted the Fe_3_O_4_ to the desired location. It is promising as one kind of carrier capable of delivering drugs to the body, that is biodegradable and simple to synthesize [65]. The different types of functional groups it included, carboxyl groups (-COOH) and carbodiimides (N=C=N), could be grafted on the particle surface. Chemotherapeutic drugs could then be conjugated to these functional groups. However, because drugs were only loaded on their surface, magnetic composites have been found to release their drug load soon after injection into the bloodstream (burst effect). The anti-neoplastic agents then failed to reach their therapeutic levels at the desired site [66]. In order to reduce their premature drug release, biocompatible polymers such as PEG molecules, polylacticco-glycolic acid, polyethylene-co-vinylacetate, and polyvinylpyrrolidone, which had been used as coating materials in aqueous solutions [67], were usually used to coat the metal cores.

Through surface modification, these coatings protected the magnetic core and allowed drug binding by forming covalent bonds, adsorption, or entrapment on the particle surface. For example, coating Fe_3_O_4_ with the cross-linked polymer, poly(ethylene glycol)-co-fumarate, caused a significant reduction in premature release (21%) compared to the non-coated magnetic particles [68]. As the purpose was increasing its targeting abilities, it improved the surface ability of the magnetic composites, which could be crafted with targeting molecules such as folic acid, RGD, proteins, transferrin, hyaluronic acid, etc. [69]. Several in vitro studies showed no, or low cytotoxic effects of Fe_3_O_4_ on cell cultures. However, others showed controversial toxicity patterns of the particle itself, from negative to positive toxicity, in several preclinical animal models in the form of surface molecule exchange or charged transfers. Generally, the toxicity of Fe_3_O_4_ depended on size, dose, surface coating, and species. The effect of exposing Fe_3_O_4_ particles to ultrasound (1 MHz and intensity of 2 W/cm^2^) on the viability of cancer cells was investigated. The Higher concentrations of Fe_3_O_4_ (above 100 mg/mL) behave as sonosensitizers generating ROS (synergic effect). Coating Fe_3_O_4_ with sonosensitizers, e.g., titanium dioxide (TiO_2_), is a promising technique for enhancing ultrasound-assisted stimulation by inducing the formation of ROS, including hydrogen peroxide and super-oxides. There is also one report by Shen et al. that investigated the cytotoxicity of DOX-loaded titanium dioxide-encapsulated Fe_3_O_4_ (Fe_3_O_4_-TiO_2_) coupled with ultrasound (1 MHz, 1 W/cm^2^). This study reported that the magnetic-based heterogeneous structure produced ROS following exposure to ultrasound. The incubation of cancer cells with Fe_3_O_4_-TiO_2_-DOX followed by sonication showed higher toxicities compared to the treatment with free DOX or Fe_3_O_4_-TiO_2_-DOX alone [70].

## 4. The Fabrication of Magnetic Metal–Organic-Frameworks (MMOF) and Related Biological Applications

Magnetism was introduced as a guest, while the framework remained magnetic properties, in order to achieve cellular transformations and signal transduction pathways in different biological applications. It was observed that living systems were proved and utilized in a series of synergistic biocompatible tissues such as proteins and amino acids, and these simultaneously processed cascade reactions in which they are used for precise localization and spatial ordering in catalytic sites [71,72]. This conveyed that the MMOF structure was modified and constructed by using biocompatible molecules. In these strategies, one of the main attributes of MOF was magnetism, which can be implemented by incorporating magnetic moment carriers such as magnetic metals or shell organic surfactant, or both of them together. These kinds of MOF composites were discriminated, like MMOF itself, and were not enough to render a material. Magnetic features such as magnetism are a cooperative phenomenon and thus require some kind of exchange between the moment carriers [73]. So far, the majority of MMOF were those containing paramagnetic metal centers and in particular, the first-row transition metals (V, Cr, Mn, Fe, Co, Ni, and Cu). These metals may exist in different oxidation states and allowed variations of the two important parameters, spin quantum number and magnetic anisotropy. These MMOF composites are of extreme interest for biomedical purposes because they can carry and deliver specific drugs in biological systems [74]. These applications are enabled by the synergy between MOFs and magnetic particles in the MMOF. Indeed, the high surface area of MMOF allowed the release of pharmaceuticals, while the magnetic particles provide sensitivity to a magnetic field, as shown in Figure 5.

The first attempts to design a biomedical device by coupling magnetically active species with MOF have been characterized and have proposed a synthesis strategy for generating new material by adding the desired magnetic particles to the MOF reaction mixture during the synthesized processes. As exhibited in experiments, a multimodal encapsulation product was obtained by adding quantum dots (QDs) along with the magnetic particles, and observing the fluorescent spheres with an optical microscope [75]. The produced MMOF composites were exposed to an alternating magnetic field; this dynamic stimulus induced localized heat procedures. This has been well-exploited for different applications, including medical treatments [76]. In the case of Cu_3_(btc)_2_ (HKUST-1, btc = benzene tricarboxylic acid) composites, the thermally triggered release of ibuprofen was evaluated; the release rate increased from 4.4 nmol s^−1^ at 20 °C to 6.6 nmol s^−1^ at 40 °C. Precise tuning of the drug release in an MOF composite was controlled by an external stimulus. Due to the natural magnetic properties of Fe_3_O_4_, integrated into the frameworks, the resulting material did not show any residual magnetization. For drug delivery systems, this was a fundamental feature that must be carefully considered, because low residual magnetization prevents uncontrolled agglomeration, reducing the chance of inducing fatal blood vessel occlusions [77]. The drug release of a set of aluminum and copper-based MOFs were studied under an alternating magnetic field [78]. Detailed information about MMOF, based on the presence of Fe_3_O_4_ particles, was freshly produced by a co-precipitation method. This method illustrated the crystalline and morphological features, similar to the original MOFs structures. The drug used for proof of concept was nimesulide, an anti-inflammatory that is also used for pancreatic cancer treatments.

In addition, the MMOF was prepared in a two-step procedure; magnetite cores were produced by co-precipitation of Fe (III) and Fe (II) salts in a basic aqueous solution first, and included in the solvothermal synthesis of the copper-doped full composites. The amount of absorbed drug can be measured after drugs loading. The magnetic composites’ surface area dramatically decreased by 95%, and an outstanding loading of up to 0.201g of drug per gram of material was achieved. It was claimed that the drug occupied almost every available space inside the framework [79]. MMOF showed a quick response to the magnetic field with a commercial magnet. As seen in Figure 6, it was mentioned in Chen [80] that magnetic multienzyme nanosystems have been prepared via co-precipitation of enzymes and MOF precursors in the presence of magnetic Fe_3_O_4_ particles. The spatial co-localization of two enzymes was achieved using a layer-by-layer positional assembly strategy. Glucose oxidase (GOx) and horseradish peroxidase (HRP) were used as the model enzymes for cascade biocatalysis. By controlling the spatial positions of enzymes, three kinds of bienzyme nanosystems were prepared; GOx and HRP- containing layers were in close proximity, either encapsulated in the HKUST-1 inner layer, immobilized on the HKUST-1 outer shell, or randomly distributed in the MMOFs’ layers. The highest activity was observed at less-alkaline atmospheres (pH = 6, 20 °C). Through the combination, the enzymes conjugations exhibit remarkably high operational stability compared to the free enzymes. It provided a facile and versatile approach to spatially organized multienzyme systems with well-defined MMOF structures. So, MMOF was treated as a scaffold for spatial co-location and the positional assembly of multienzyme systems, enabling enhanced cascade bio-catalysis and further applications. Meanwhile, the obtained MMOF structures (γ-Fe_2_O_3_@MOFs) remained highly crystalline, porous, and thus accessible to the guest molecules. Additionally, it demonstrated the potential of MMOF for controlled drug delivery [81]. It was clearly shown that the integration of this superparamagnetic property into MOFs endowed the resulting materials with excellent magnetic performance for controllable magnetic separation and drug release behavior too.

## 5. Conclusions

Here, we presented a detailed overview of the role that metal-based magnetic composites as one kind of high-efficient functional agents play in biological applications. Especially in constructing procedures, the metal elements were mixed in one-pot or hot- injecting approaches and iron oxides and heterogeneous structures were well-fabricated. We presented new conceptions for biological applications and further investigation for cancer therapies. We offered one possibility for medical clinical application and new drug delivery systematic fabrications. Particularly, the heterogeneous magnetic materials can be treated as photosensitizers and sonosensitizers, which fixes the whole system as one flexible substrate. This should be the preferred option in integration towards cancer cells. This would not only prevent sedimentation and promote candidates recycling, but also alleviate the cost of reduced mass transfer between the target cancer issues and reactants. In the long run, it is reliable for the collective optimization of metal-based magnetic photosensitizers or sonosensitizers, reaction media and reactors, and their practical application for cancer therapy valorization. In spite of the promised future research field, the developed multifunctional magnetic materials and their systems for SDT applications are limited. By combining with US techniques in SDT and integrating with photosensitive effects, the desired requirements could be further designed and advance towards dual-mode integrations.

Till now, great progresses has been achieved in the design and synthesis of magnetic materials, which can be incorporated with ultrasonic applications. Thus, the tailoring of active functional materials would greatly contribute to achieving a high-efficiency ultrasound conversion system. Furthermore, compared with existing materials, the question of how to utilize magnetic materials is the most common difficulty. By utilizing magnetic materials would expand the range of metal element fabrications and also enhance the function toward cancer therapy, once completely modified by the functional groups. Among cancer therapies, the medium condition was the special requirement for metal elements, and its acquirement should be an inert candidate without any changes in its properties’ exhibition. Integrating with US treatment, it generally induced nonselective and noninvasive strategies for cancer treatment. Especially for the metal platform, the chemical stability was easy to influence, and damages in whole constructions appeared. The fixed US agents should be taken into account for real applications and conform to requirements.

## Figures and Tables

**Figure 1 ijms-22-10461-f001:**
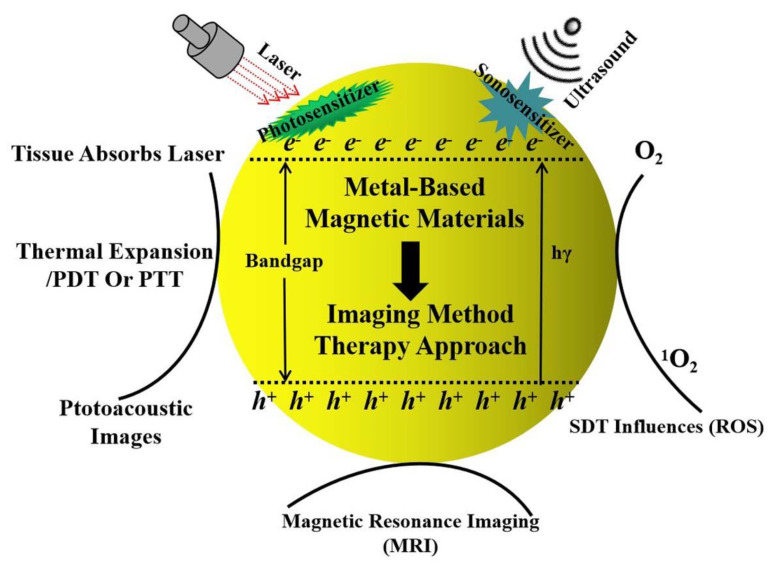
Scheme of metal-based magnetic materials occurring through different further applications due to mechanisms formed in real treatment.

**Figure 2 ijms-22-10461-f002:**
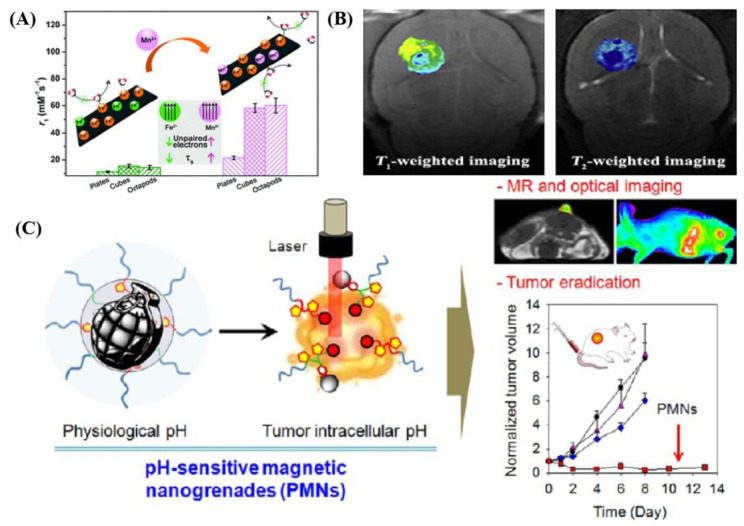
Metal-based magnetic composites as a functional candidate in (**A**) surface manganese substitution in magnetite crystals enhancing T_1_ contrast ability by increasing electron spin relaxation; (**B**) T_1_–T_2_ dual-modal MRI of brain gliomas using PEGylated Gd-doped Fe_3_O_4_; (**C**) Schematic illustration of pH-sensitive self-assembled structure for MRI and treatment of resistant heterogeneous tumors. These are reprinted with permission from Zhao et al. (2018, RSC), Xiao et al. (2014, Elsevier), and Ling et al. (2014, ACS), respectively.

**Figure 3 ijms-22-10461-f003:**
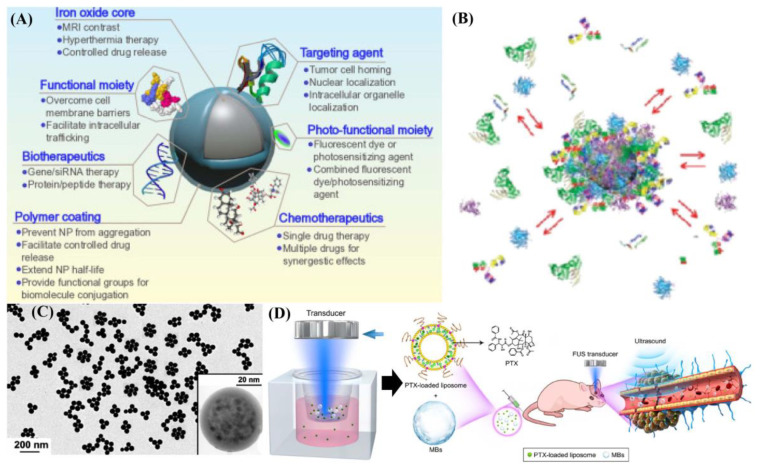
(**A**) Schematic illustration of a full-suite theranostic NP. The magnetite core serves as an MRI contrast agent and heat source for magnetic hyperthermia, and a polymer coating increases biocompatibility, mitigates RES uptake, and allows for facile functionalization with chemotherapeutic, biotherapeutic, and optical enhancement, and targeting moieties. (**B**) Schematic drawing of the structure of protein-nanoparticles in blood plasma confirming the existence of various protein binding (e.g., an outer weakly interacting layer of protein (full red arrows)) (**C**) Mono-disperse superparamagnetic nano spherical composites with a core containing metallic α-Fe nanocrystals dispersed in a silica matrix, and a shell only containing silica. (**D**) The effect of FUs with MBs on the permeability of the in vitro BBB mode and schematic illustration outlining the delivery of PTX-lIPO using FUs exposure in the presence of circulating MBs. These are reprinted with permission from Mahmoudi et al. (2011, Elsevier), Shen et al. (2017, Dove Press), Tartaj et al. (2003, Elsevier), and Revia et al. (2015, Elsevier), respectively.

**Figure 4 ijms-22-10461-f004:**
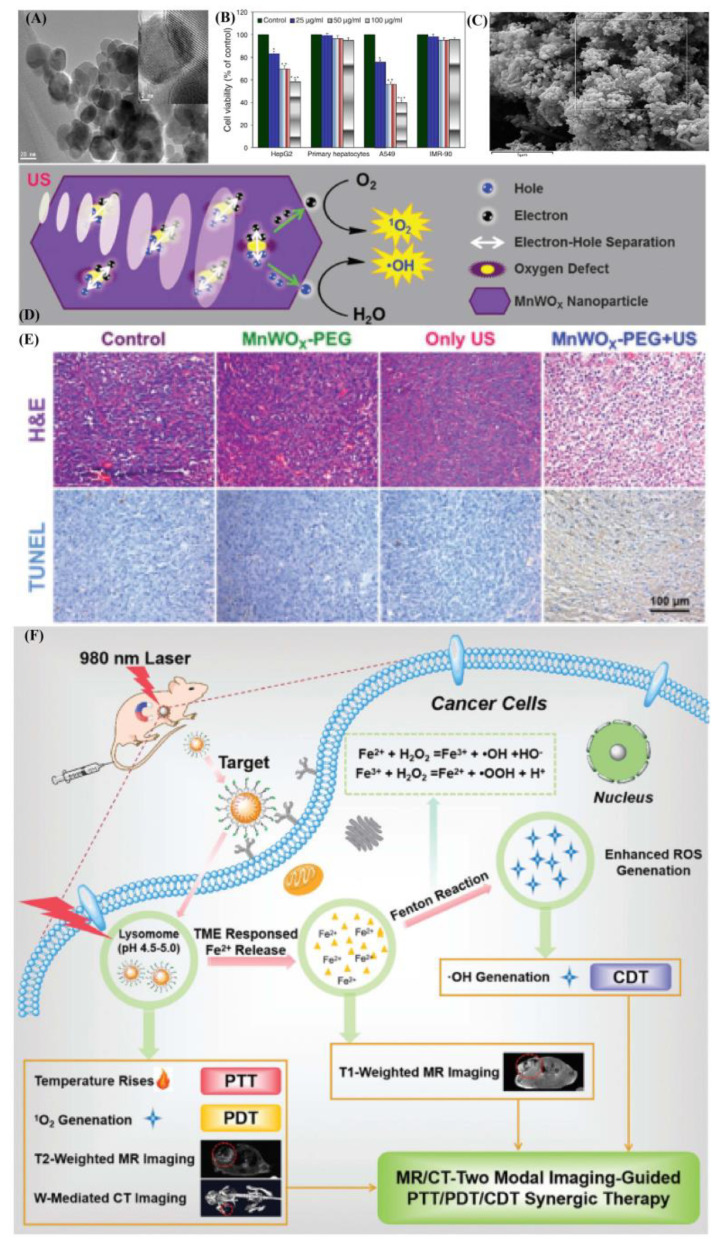
Metal-based magnetic composites as functional hard template in fabricating typical materials. (**A**,**B**) Structural information of pure phased Fe_3_O_4_ and biocompatibility were tested in normal and cancer cell lines. Effect of Fe_3_O_4_ on the viability of cancer cells and two types of normal cells; (**C**) FE-SEM image structural information of Fe_3_O_4_@C core-shell particles; (**D**,**E**) The proposed mechanism of ROS generation by MnWO_X_-PEG under ultrasound irradiation. Microscopy images of H&E and TUNEL-stained tumor slices; (**F**) Illustration of the fabrication of FeWOx–PEG–RGD NPs for (T_2_/T_1_-weighted) MR/CT dual-modal imaging-guided PTT/PDT/CDT synergistic therapy. These are reprinted with permission from Ahamed et al. (**A**,**B**: 2013, Elsevier), Bakhshi et al. (**C**: 2020, Elsevier), Gong et al. (**D**,**E**: 2019, Wiley) and Cheng et al. (**F**: 2021, RSC).

**Figure 5 ijms-22-10461-f005:**
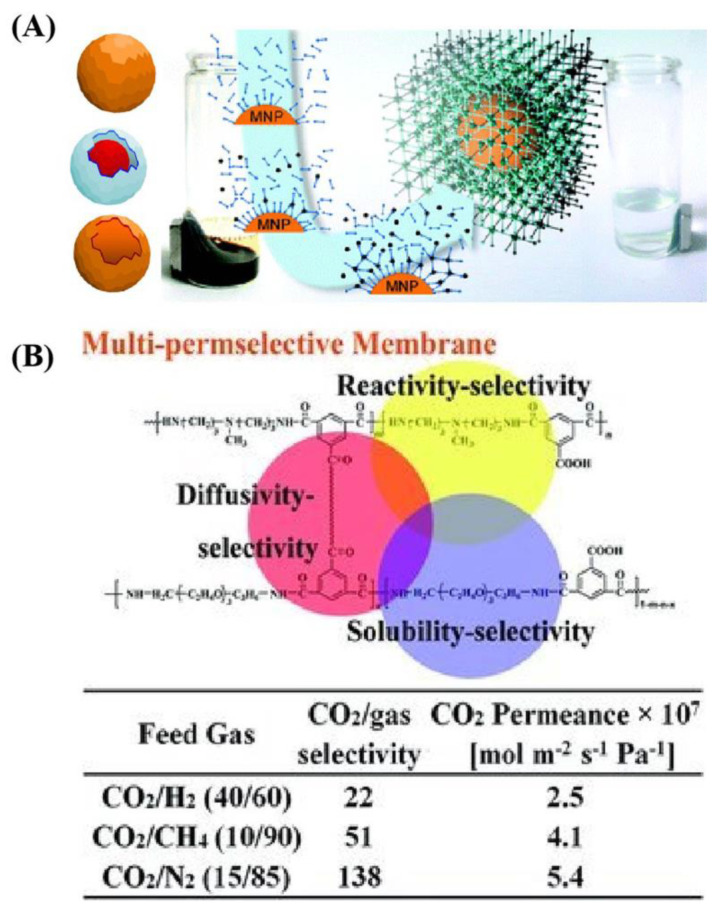
(**A**) Schematic illustration of MMOF by using different structural magnetic composites; (**B**) High-performance membranes with multi-permselectivity for CO_2_ separation. These are reprinted with permission from Lohe et al. (2011, RSC) and Li et al. (2012, Wiley).

**Figure 6 ijms-22-10461-f006:**
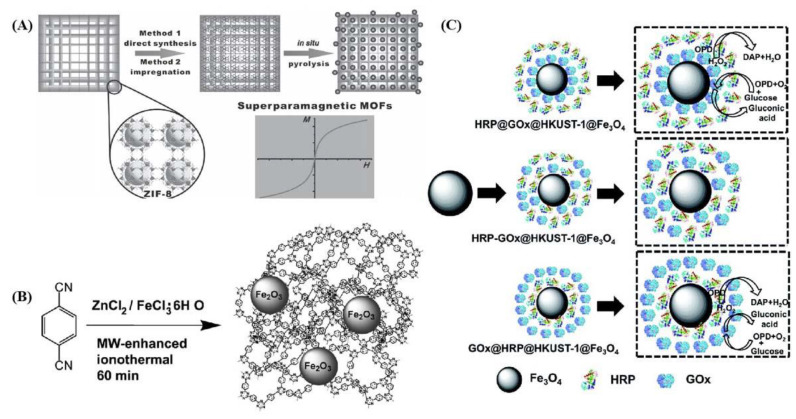
Schematic illustration of MMOF synthesized procedures and related applications. (**A**) Preparation of superparamagnetic metal–organic frameworks of γ-Fe_2_O_3_@ZIF-8; (**B**) CTF/Fe_2_O_3_ composite synthesized by the microwave-enhanced high-temperature ionothermal method, (CTF = covalent triazine-based framework); (**C**) Three different strategies for the positional assembly and spatial co-localization of glucose oxidase and horseradish peroxidase. These are reprinted with permission from Wu et al. (2014, Wiley), Zhang et al. (2011, Elsevier), and Chen et al. (2017, RSC), respectively.

## Data Availability

Not applicable.

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
