# Peer review of "Recent Advances in Metal-Based Magnetic Composites as High-Efficiency Candidates for Ultrasound-Assisted Effects in Cancer Therapy"

_ijms, 2021, doi:10.3390/ijms221910461_

Round 1
Reviewer 1 Report
The concept of describing the recent advances in metal-based magnetic composites as high-efficiency candidate for ultrasound assisted effect in
cancer therapy is an interesting topic for the readers however authors are not summarizing the topic in a good way.
-Language of the manuscript needs to be improved. Abstract and conclusion are very poor in writing.
The abstract is not concise and descriptive of the concept in the title of the manuscript. Rather than focusing on the metal based materials authors should conclude why metal-based magnetic materials are receiving attention and which material is the focus of attention and why and what is the agenda of this review article.
As the introduction section is very weak, rather than describing the background and need of the topic it seems like it is just a compilation of information from different sources from a variety of sources.
Figure. 1 needs to be modified or clarity in the topic is neeeded. As author is mainly talking about iron oxide then figure 1 should be modified for iron oxide only.
The explanation for the mechanism of MRI is not sufficient.
Section 2 needs a conclusion before jumping towards US integrated therapy to highlight the need for integration.
A separate section is needed to explain the mechanism and limitation of US-based therapy.
Section for integrated therapy would need a working principle of integration therapy in cancer treatment which will be followed with the synthesis methods (which have already been discussed).
The concept of PAT should be described at the end of this section to show additional advantages of integrated therapy.
Reviewer 2 Report
Dear Editor,
The authors claim to present a review of the use of metal-based magnetic composites in ultrasound cancer therapies. The two first parts of the article are quite unrelatedly written and it is not clear what the authors wanted to show. The part related to ultrasound therapy is clearer. In general, the advantages of the use of metal-based magnetic composites in ultrasound therapy have not been supported by data. Most of the citations are not given in full, for example missing pages, volume number or year. This article is not a complete article and has many shortcomings. In my opinion, this manuscript is not acceptable in this form.
- On line 41-42 was written “1) Iron based chalcogenides such as Fe3O4, Fe2O3, FeO, FeS, FeS2, FeSe, FeSe…”, but iron oxides are not chalcogenides. Please, correct.
- The Introduction part should be rewrite. The presented information is not systematized and it is not clear what is the purpose of the manuscript. It is not clear what the Figure 1 presented.
- Line 149: “As showed in Figure 2, according to the previous studies, …”. Please, give references.
- Please, give more information for the “three different regimes” mention on line169 and their influence on the magnetic property of Fe3O4.
- The part 2 entitled “2. Superparamagnetic iron oxides nanoparticles (SPIOs) as an effective agent in magnetic resonance imaging (MRI) utilization” does not provide detailed information on why magnetite or other SPIOs can be used as a contrast agent. What are its advantages and disadvantages.
- Figure 2 presented the results for doped Fe3O4 obtained by the other authors. Please, give more information for the results and the influence of particle size and shape and their magnetic properties on MRI.
- Line 307: “As illustrated in previous studies,…”. Please, give references. Similar for line 330.
- Please, give information for toxicity of NiFe2O4 and manganese oxides.
- Line 350: “Magnetism can also be introduced as guests while the framework remained non-magnetic properties.”. What do you want to say?
- What does “HKUST-1 composite” mean?
Reviewer 3 Report
The review article entitled: “Recent Advances in Metal-Based Magnetic Composites as High-Efficiency Candidate for Ultrasound Assisted Effect in Cancer Therapy” describes various magnetic composited which,
The entire article is very chaotic and hard to read. The Authors meant to describe recent accomplishments in ultrasound-assisted cancer therapy. If, in fact, the article would describe this area of magnetic composites application, the review could be of great scientific interest. I believe the Authors need to rewrite the article, giving it a better structure and more up-to-date references. In this form, the article is not publishable.
Some specific remarks for future help:
The title does not reflect the article at all.
Keywords: it seems the Authors copied various words from the title, not really considering their appropriate use. What would you mean by keyword: candidate? Moreover, keywords such as metal or tissues are not specific enough.
Introduction: Introduction presents types of magnetic particles and composites used for various biomedical purposes. Please note that (1) chalcogenides usually do not refer to oxides; (2) functionalization does not make magnetic particle a composite, (3) introduction does not cover any aspect of ultrasound-assisted therapy, (4) much attention is given to synthesis, which is not further reflected in the main text and (5) finally, the authors basically change the area of the review article (line 104) to magnetite only.
Round 2
Reviewer 1 Report
- I appreciate authors carefully revised the abstract describing “metal-based magnetic materials are receiving attention and highlighting which material is the focus of attention and why. Authors need to concise the introduction section in a similar way.
Reviewer 2 Report
Dear Editor,
The authors claim to present a review of the use of metal-based magnetic composites in ultrasound cancer therapies. They should change the title, because they did not report for any magnetic composites. They improved their manuscript, but still some unrelatedly written and as I reported before in the two first parts of the article it is not clear what the authors wanted to show. The part related to ultrasound therapy is clearer. Sometimes they use “The detailed information” or “The study showed” or “It reported” without give the references. Why the magnetic particles should be superparamagnetic to be used as contrast agent in MRI? Figure 2 presented the results for doped Fe3O4 obtained by the other authors. Please, give more information for the results and the influence of particle size and shape and their magnetic properties on MRI. What size and shape is the best for MRI application.
Reviewer 3 Report
The Authors took some suggestions under consideration, nevertheless, I believe the article in such a form cannot be accepted for publication in IJMS. The article is still very chaotic, moreover, practically all the added parts are poorly written in terms of english language and spelling.
Author Response
We highly appreciate your comments. Hopefully, our reply as listed below may address your concerns.
1-The Authors took some suggestions under consideration, nevertheless, I believe the article in such a form cannot be accepted for publication in IJMS. The article is still very chaotic, moreover, practically all the added parts are poorly written in terms of English language and spelling.
-Reply: Thanks for reviewer’s comment. As mentioned in comment, we have reviewed the contents and reorganized the contents as much as possible, the related and updated contents as much as possible. Please check it.
Round 3
Reviewer 2 Report
Dear Editor,
The authors answer to my questions and did the appropriate changes in the manuscript. The manuscript has been improved and is more clearer. It may be accepted for publication in the present form.
Best regards
Author Response
Thanks so much for reviewer's comment, we will do our best and make this review manuscript acceptable as much as possible. Really appreciate for yours kind comments and suggestions while we did the corrections.
Thanks again.
Reviewer 3 Report
The entire article needs extensive editing in terms of English grammar and punctuation.
Just at the example of abstract (but be aware that this problem concerns the entire article, not just the abstract), there are plenty of mistakes in every single sentence:
- SENTENCE 1: First sentence is hard to read and needs clarification. Line 11: What does it mean well used? In that term, the word well is wrongly used. Line 12: in which (1) in which properties? (grammar) (2) unnecessary comma after the word “which”. Try dividing it into 2 sentences.
- SENTENCE 2: Line 13: comma before “including”, line 14: should be “imaging AND monitoring”. Line 14 further: “… the fabrication of metal based magnetic materials can be doped with….” Fabrication can be doped? (grammar); metal-based (with a hyphen). Line 16: word “research” does not have a plural form in this context.
- SENTENCE 3: Line 16: “it can lead one” (1) what can lead? (2) to what? (grammar). What does it mean human activities?
- SENTENCE 4: “As appeared in property characterization…”, (1) grammar, (2) one property? Line 19: influenced or influences? Decide which time and voice you want to use in the abstract.
- SENTENCE 5. Line 19: “Toward to” – wrong preposition use. Line 20 “metal based” – hyphen is missing, “one appropriate” – the only one? line 21 should be “to the ability to penetrate”
- SENTENCE 6: the verb is missing in that sentence. Rewrite. Moreover, "influences further", not "furtherly".
- SENTENCE 7: Line 24: “At the same time”; “by combined with cancer cells” -grammar; imaging AND monitoring; unnecessary comma after “areas”; “cure process”?;
- SENTENCE 8: line 27: should be: “of various types”; should be “of the magnetic materials”; What does it mean “actions of their applications”? should be “in the magnetic resonance imaging”
- SENTENCE 9: Line 28: What exhibits the mechanism? This review?; the ultrasound
- SENTENCE 10: “pure phased” or” phase-pure”? “…with crucial roles in cell signaling, tissue imaging, expression in cancer cells, and their high …”
- SENTENCE 11: the interactions may be used for targeted drug delivery? Grammar. , “its surface” – which surface? From the context of the sentence, it appears that you mean the interactions’ surface – again, grammar.
- SENTENCE 12: hyphen missing and unnecessary comma.
Moreover, as stated earlier, the title is not clear. What does it mean “Candidate for Ultrasound Assisted Effect” in the context of this statement? Keywords: as mentioned in the previous review, some keywords are not informative enough. What would you mean by “heterogeneous” or “ultrasound”?
As my job is not editing this article but reviewing it, I will not point out its every mistake. Please, revise, edit and improve the entire paper before submitting it again.
